 

# Message from the new Executive Editor

Tim Fessenden ⓘ

It is my pleasure to introduce myself to readers, authors, and reviewers as the new Executive Editor of Life Science Alliance (LSA). Together with Sarita Hebbar, who joined the journal as Scientific Editor in January, I am committed to expand on the successes of this innovative journal to the benefit of authors and readers worldwide. Here I share my view of this exciting role, which lies at the heart of the biology and biomedical research enterprise.

**DOI** https://doi.org/10.26508/lsa.202503362 | Received 17 April 2025 | Accepted 17 April 2025 | Published online 22 April 2025

What do editors do? Postdocs and graduate students who are mapping their career prospects frequently ask me to describe this role. My colleagues and friends in faculty positions are often just as curious about life on this side of the manuscript submission portal. Describing this job is very simple in granular terms. I can easily tick through the daily, piecemeal tasks of reading manuscripts, communicating recommendations to colleagues, commenting on their views, or digging into peer reviewer comments. I find it far more difficult to zoom out and convey what I do on the scale of months and years. What do so many manuscript evaluations, peer review summaries, and decision letters amount to? What exactly do all those tasks build? The simple answer is journals. By identifying and promoting good science, treating authors with decency and respect, and ensuring quality of peer review, among many other activities, editors nurture the reputation and quality of journals.

But stepping back from the mechanics of publishing, we editors have a front row seat for the advance and fine-tuning of the state of knowledge in biology. Viewed across generations, the steady accumulation of experimental and observational evidence in biology takes shape as a bustling, self-correcting, constantly improving understanding of the living world. Through our work at journals, editors facilitate collective decisions at the core of this evolving state of knowledge. Put another way, editors help biology think.

Of course, this means that editors are not passive observers. Together with authors, reviewers, and, indirectly, readers, editors participate in generating the record of what will count as biology and biomedicine. The essential features of this role are partnership and trust. Editors are not able to compel researchers to read their journals or to cite the articles within. We rely, by default, on the expertise of reviewers whose integrity and faithful assessment we must trust. Nor do editors govern whether and when authors will submit their hard-earned research manuscripts. Through these relationships, and on a bedrock of trust, editors serve as shepherds, curators, and advocates for the publication record.

In the lush ecosystem of biology journals, Life Science Alliance remains wholly unique. The journal began and remains an experiment in scientific publishing and the fully open access journal model. Leaders at three nonprofit, academic publishers — Cold Spring Harbor Press, EMBO Press, and Rockefeller University Press — launched LSA as a jointly operated journal in 2017. Editor communication between LSA and 10 partner journals means that LSA benefits directly from the experienced and rigorous editorial practices at partner presses. If a manuscript under consideration at a partner journal is declined, whether at the pre-review/editorial stage or at the post-review stage, LSA editors are able to make an informed offer to transfer. For post-review manuscripts, this is most often an offer to publish the manuscript in question. I participated in these consultations in my prior role as a Scientific Editor at the Journal of Cell Biology, where I began my career in editing in 2021. Beyond these innovations in transfers across publishers, LSA serves authors by swift decisions and limiting manuscripts to a single round of major revisions. Finally, LSA participates in the Review Commons consortium which has proven to be a successful innovation in journal-agnostic peer review.

I am convinced that editors do an essential part of their job simply by meeting with researchers and discussing their work. This includes at LSA, where I am excited to travel to meetings and institutions and I am accessible to offer feedback and advice on research manuscripts at all stages. Hand in hand with accessibility is my commitment to demystify the editorial process, as far as I am able. Shedding light on the role of editors will benefit authors and readers of LSA as well as those curious about careers in editing. Discretion and confidentiality are non-optional features of editorial decision-making, but in my experience, these are easily separable from open discussions about my stewardship of the journal and scientific communication for the benefit of all.

While I am honored to be appointed to the role of Executive Editor, and I am full of excitement and hope for LSA, I step into this role in a time of unprecedented challenges to academic research in the United States. The weekly, if not daily, news of cuts to the NIH and NSF, as well as actions aimed directly at academic institutions,

Life Science Alliance LLC, New York, NY, USA

Correspondence: t.fessenden@life-science-alliance.org

is at best demoralizing and at worst an existential threat to the fantastic research infrastructure in this country. These are threats to academic freedom in the broadest sense, coupled with grievous reductions to the largest research system in the world. This moment has been aptly compared to the Red Scare of the 1950's, although that episode did not attempt to control and curtail biological and biomedical research.

I am just as alarmed and concerned by these changes as my many friends and colleagues in research positions. Like many others, I am all the more aware of the value of research, the impressive feats of US research infrastructure, and the strength of its public support now that it is being threatened and dismantled before our eyes. This awareness and appreciation should serve as a source of hope for a future in which scientific research enjoys stronger public understanding and support in the US. Here I have optimism, a sentiment in very short supply. Of course optimism does not by itself change the policies of the US government. But without the conviction that things can improve, they certainly will not. In the near term, beyond advocating for change in political and legal realms, perhaps optimism itself is a broadly distributed and

achievable goal. Our advocacy, in all forms, for the profound value of investigating the natural world, must itself become a source of optimism. That is more than enough reason to meet this moment with action: to create the very conditions that engender optimism in scientists of all career stages.

In closing, I commit to ensuring that LSA remains a high-quality and low-friction outlet for exciting advances spanning several research areas. In the midst of systemic and concerning changes in biological and biomedical research, LSA strives to lower barriers to publication and meet authors where they are. I look forward to serving authors, readers, and reviewers at LSA with humility, openness, and optimism.

Please contact me with your feedback or concerns at t.fessenden@ life-science-alliance.org.

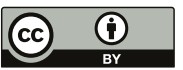

