## [Reviewer comments · Life Science Alliance]

Message from the new Executive Editor

Author information redacted

DOI: <https://doi.org/10.26508/lsa.202503362>

Corresponding author(s): Author information redacted

Review Timeline:

Submission Date:

2025-04-17

Accepted:

2025-04-17

Scientific Editor: Sarita Hebbar

Transaction Report:

April 17, 2025

RE: Life Science Alliance Manuscript #LSA-2025-03362

Tim Fessenden
Life Science Alliance LLC
The Rockefeller University Press
950 Third Avenue
2nd Floor
New York, NY 10022

Hi Tim, In case you wondered what the decision would be, here it is!

Thank you for submitting your Editorial entitled "Message from the new Executive Editor". It is a pleasure to let you know that your manuscript is now accepted for publication in Life Science Alliance. Congratulations on this interesting work.

DISTRIBUTION OF MATERIALS:

Again, congratulations on a very nice paper. I hope you found the review process to be constructive and are pleased with how the manuscript was handled editorially. We look forward to future exciting submissions from your lab.

Sincerely,

Sarita Hebbar, PhD
Scientific Editor
Life Science Alliance
<http://www.lsajournal.org>

April 17, 2025

RE: Life Science Alliance Manuscript #LSA-2025-03362

Tim Fessenden
Life Science Alliance LLC
The Rockefeller University Press
950 Third Avenue
2nd Floor
New York, NY 10022

Hi Tim, In case you wondered what the decision would be, here it is!

Thank you for submitting your Editorial entitled "Message from the new Executive Editor". It is a pleasure to let you know that your manuscript is now accepted for publication in Life Science Alliance. Congratulations on this interesting work.

DISTRIBUTION OF MATERIALS:

Again, congratulations on a very nice paper. I hope you found the review process to be constructive and are pleased with how the manuscript was handled editorially. We look forward to future exciting submissions from your lab.

Sincerely,

Sarita Hebbar, PhD
Scientific Editor
Life Science Alliance
<http://www.lsajournal.org>